# The Role of Aquaporins in Epileptogenesis—A Systematic Review

**DOI:** 10.3390/ijms241511923

**Published:** 2023-07-25

**Authors:** Lapo Bonosi, Umberto Emanuele Benigno, Sofia Musso, Kevin Giardina, Rosa Maria Gerardi, Lara Brunasso, Roberta Costanzo, Federica Paolini, Felice Buscemi, Chiara Avallone, Vincenzo Gulino, Domenico Gerardo Iacopino, Rosario Maugeri

**Affiliations:** Neurosurgical Clinic, AOUP “Paolo Giaccone”, Post Graduate Residency Program in Neurologic Surgery, Department of Biomedicine Neurosciences and Advanced Diagnostics, School of Medicine, University of Palermo, 90127 Palermo, Italy; lapo.bonosi@gmail.com (L.B.); umberto.emanuele.benigno@gmail.com (U.E.B.); sofiamusso.sm@gmail.com (S.M.); kevingiardina97@gmail.com (K.G.); rosamariagerardimd@gmail.com (R.M.G.); brunassolara@gmail.com (L.B.); robertacostanzo3@gmail.com (R.C.); federicapaolini94@gmail.com (F.P.); felice.buscemi96@gmail.com (F.B.); avallonechiara97@gmail.com (C.A.); vincegulino1@gmail.com (V.G.); gerardo.iacopino@gmail.com (D.G.I.)

**Keywords:** aquaporins, AQP4, epilepsy, seizure, ion homeostasis, pathophysiology

## Abstract

Aquaporins (AQPs) are a family of membrane proteins involved in the transport of water and ions across cell membranes. AQPs have been shown to be implicated in various physiological and pathological processes in the brain, including water homeostasis, cell migration, and inflammation, among others. Epileptogenesis is a complex and multifactorial process that involves alterations in the structure and function of neuronal networks. Recent evidence suggests that AQPs may also play a role in the pathogenesis of epilepsy. In animal models of epilepsy, AQPs have been shown to be upregulated in regions of the brain that are involved in seizure generation, suggesting that they may contribute to the hyperexcitability of neuronal networks. Moreover, genetic studies have identified mutations in AQP genes associated with an increased risk of developing epilepsy. Our review aims to investigate the role of AQPs in epilepsy and seizure onset from a pathophysiological point of view, pointing out the potential molecular mechanism and their clinical implications.

## 1. Introduction

Aquaporins (AQPs) are a group of specific transmembrane channel proteins responsible for facilitating the movement of water across cell membranes [1,2]. Historically, the archetype of this protein family is the major intrinsic protein (MIP) found in mammalian lens fibers (also known as AQP0). After it was shown that homologs of MIP possessed water-sensitive channel protein activities, they were generically renamed aquaporins. Later, with increasingly better functional characterization of these homologs, it was discovered how some were exclusively permeable to water (taking the name aquaporins), while others were also permeable to other molecules such as glycerol, thus being named aquaglyceroporins [3]. Until now, 13 AQPs have been identified in the human body and are predominantly found in epithelial tissues requiring a rapid flow of water (Table 1). Among them, AQP1, AQP9, and AQP4 have been described in the central nervous system (CNS), while different aquaglyceroporins have been described in the adipose tissue, intestine, liver, kidney, and endocrine pancreas and seem to be related to obesity and type 2 diabetes [4,5].

As regards AQPs found in CNS, AQP1 is normally expressed in the apical pole of the choroid plexus of epithelial cells and can also be well represented in astrocytes in some pathological conditions, such as brain edema and glial tumors. Its role is to act as a water channel and as a cGMP-gated ion channel, thus maintaining the normal water and ion balance and allowing the secretion of cerebrospinal fluid [6]. In this regard, Rauen et al. [7] hypothesized that these aquaporins may be regulated by vasopressin V1a receptors. Moreover, modified AQP1 expression may determine choroid plexus epithelial cell disfunction, leading to alterations in cerebrospinal fluid production and thus affecting the water homeostasis. Interestingly, in their study, Srisook and colleagues demonstrated the role of cerebral malaria (CM) in inducing apoptotic changes to choroid plexus epithelial cells, to define AQP1 as a potential target to reduce brain edema associated with CM [8]. On the other hand, AQP9 represents the only aquaglyceroprotein in the brain. It is involved in energy metabolism, transporting both water and other substances, such as glycerol, amino acids, and lactate; recently, AQP9 expression has been observed in neurons of the brainstem and midbrain, predominantly in the catecholaminergic neurons [9].

Aberrant changes of AQP9 have been related to brain disorders: for instance, Opdal et al. identified genetic variations in AQP1 and AQP9 together with the AQP4 complex as potential predisposing factors involved in sudden infant death syndrome [10,11]. Instead, the AQP4 has a widespread expression in the CNS and has proven to be related to system water and potassium homeostasis, as well as to neurotransmission. Over the years, literature has focused on the induced changes in AQP4 activity because it has been related to several CNS disorders; for instance, in brain edema, the expression of AQP4 is increased, while its reduction is related to edema improvement [12,13]; in glioblastomas and high-grade gliomas there is a loss of AQP4 polarity, which may contribute to the increased tumor cell migration, survival and spreading [14]; the neuromyelitis optica (NMO) is characterized by the presence of circulating IgG1 antibodies against the AQP4 [15]; in Alzheimer’s disease, AQP4 deficiency is related to an alteration of the glutamate transport pathway and to K+ ion homeostasis, leading to an impaired memory [16]. It therefore emerges that AQP4 expression can influence several aspects of CNS; among them, epilepsy also seems to be related to AQP dysfunction, because of the role in water and ion balance [17,18].

Special attention among the various isoforms of AQPs certainly deserves AQP4. Historically, it was discovered in 1986, and through various experimental studies, its major expression was observed in the cerebellum and gray matter of the spinal cord, thus representing the most widespread isoform of AQP in the CNS [19]. From a molecular point of view, it consists of six alpha helices crossing the membrane, connected by five loops. Each subunit has two distinct domains: the transmembrane domain and the cytoplasmic domain [20]. The former forms the water and ion channel, while the latter is involved in the regulation of trafficking and localization of AQP4 within the cell (particularly at the astrocytic level). Each monomer of AQP4 is joined with three other monomers to create a tetrameric functional unit that represents the true channel for the passage of water and ions [21,22]. Interestingly, AQP4 has a unique structural feature known as the “NPA motif” (asparagine-proline-alanine), which is highly conserved among the various aquaporin isoforms. The NPA motif contributes to the selectivity of AQP4 for water molecules and prevents the selective passage of ions and other solutes [2,23]. After tetramerization, AQP4 clusters on the cell membrane, driven by specific protein–protein interactions, organizing into regular and specialized arrays, thus forming the characteristic orthogonal arrays of particle (OAP). The formation of OAPs is believed to be influenced by various factors, including the presence of other proteins and molecules in the cell membrane, as well as specific signaling pathways. It is believed that the function of OAP is to optimize the efficiency of water transport, thus maintaining water balance in the CNS [24,25]. However, the functional significance of AQP4 assembly in OAP is still an area of active research.

In this scenario, many efforts are now being directed not only at AQP4 structure and function but also at the corollary of proteins that anchor and stabilize AQP4 at astrocyte endfeet [26,27]. The key protein involved in the anchoring of AQP4 to perivascular end feet astrocytes is called dystrophin, which is part of a larger protein complex called dystrophin-associated protein complex (DAPC) [28,29]. DAPC interacts with AQP4 through a protein called alpha-syntrophin. The AQP4 anchoring system is essential in the brain because physiological functioning of AQP4 channels contributes to fluid circulation and ion homeostasis between blood and brain tissue [30,31]. DAPCs also contain other associated proteins, such as beta-dystroglycan, beta-sarcoglycan, and gamma-sarcoglycan, all of which are involved in the stability and integrity of the DAPC complex. Another crucial role is played by the M23/M1 isoforms of AQP4, both presenting specific features and functions [32,33]. They are capable of interacting with the DAPC to anchor AQP4 to astrocyte end feet, helping to stabilize AQP4 localization in the perivascular end feet. Notably, the presence of both isoforms allows for functional diversity and adaptability in water transport regulation in the astrocyte end feet (Figure 1). Many other proteins participate in the AQP4 anchoring system at the level of the astrocytic perivascular end feet, determining their homeostasis and proper function. For a more detailed view, however, we consider it more appropriate to refer to other work pertaining to this specific focus [34,35,36].

Epilepsy comprises a group of extremely common disorders characterized by the recurrence of unprovoked seizures, which can be both acquired and congenital [37]. Currently, there remains limited knowledge regarding its pathogenesis and natural progression. It is estimated that approximately 30% of individuals affected by this condition do not attain effective seizure control, even after treatment with three or more antiepileptic drugs, leading to the emergence of a condition known as intractable epilepsy (IE) or drug-resistant epilepsy (DRE). Over the years, many efforts have been made to find molecular substrates implicated in the genesis of epilepsy, which can serve as a pharmaceutical target [38]. For example, some studies have demonstrated how alterations in potassium clearance and impaired astrocyte-dependent water homeostasis were correlated to epileptogenesis, and the inhibition of AQP4 expression has been related to a relief in epilepsy symptoms, leading some authors to speculate about the use of AQP4 inhibitors in its treatment [39,40,41].

The partial knowledge of epileptic pathogenesis and the presence of IE reflect problems in the therapeutic approach to the patient affected by epilepsy and emphasize the need for new pharmacological strategies. In this regard, the aim of this review is to pinpoint some of the possible molecular mechanisms of this pathology, highlighting the potential role of aquaporins in its etiopathogenesis, thus identifying promising scenarios for new pharmacological treatment.

## 2. Methods

### 2.1. Search of the Literature

We performed a broad literature search, following the PRISMA flowchart, on the PubMed database for studies investigating the role of AQPs in epileptogenesis and epilepsy-related brain disease [42]. We searched for studies published up to 4 November 2022 without backward limits, using the following MeSH: “aquaporins” AND “epilepsy” and adding one or more of the following free text terms: “pathogenesis”, “molecular mechanisms”, “dysregulation”. To avoid potential omission of relevant studies, we also manually screened the reference list of papers included and previous reviews regarding similar topics. Duplicate articles were eliminated using Microsoft Excel 16.37 (Redmond, WA, USA).

### 2.2. Study Selection

The research strategy relied on title and abstract analysis. An article’s full text was retrieved if the title and abstract met the inclusion criteria. No automatic tools were used in this phase. 

### 2.3. Inclusion Criteria

We included in our selection both animal and human studies evaluating the role of AQPs in epileptogenesis and their role in pathological conditions predisposing to developing epilepsy; only English-language studies were considered. We excluded studies focusing on the role of AQPs in brain disease not related to epileptogenesis or not evaluating the molecular mechanism of seizure onset.

### 2.4. Qualitative Data Extraction

According to the abovementioned criteria, all articles were screened and identified by three reviewers (U.E.B, S.M, and K.G). Disagreement was resolved with discussion and consensus, and when discussion failed to lead to consensus, a third researcher mediated (L.B.). The extracted qualitative data included the following: author, publication year, aim of the study, and principal findings.

## 3. Results

Data Selection and Study Chracteristics

From the literature search performed on PubMed with the abovementioned queries, we identified 322 articles, which became 284 after removal of duplicates. After exclusion for title and abstract, we obtained 91 papers to be assessed for eligibility. These were later screened for relevance, and finally, 19 were included according to our inclusion criteria and the goal of the review. Full text was available for all 20 included studies, which were included in our qualitative analysis (Figure 2).

We arbitrarily decided to subdivide the studies according to some salient features decided a priori, with the intention to make the reader understand the multiple diseases correlated with epilepsy in which AQPs are implicated and the dysregulation of their physiological homeostasis. The characteristics of included studies are summarized in Table 2.

## 4. Discussion

The Global Burden of Epilepsy Report estimates that there are about 50 million people with epilepsy in the world, and of these, up to 35% are drug-resistant [58,59,60]. If we imagine epilepsy pathogenesis like a theatrical piece where multiple molecular actors are involved, we must highlight the roles played by AQPs in mediating some of the pathological processes related to the genesis of epilepsy. As a matter of fact, these membrane proteins participated in multiple aspects of this pathology, such as inflammation, brain edema formation, ion homeostasis, and many others, as demonstrated by several studies [61,62,63,64].

For instance, brain water homeostasis plays a remarkable role in neural activity balance. In this context, ionic equilibrium and water distribution are milestones in brain tissue activity [65].

Even if physiopathology of epilepsy is mainly associated with impaired neuronal transmission, evidence has widely proven that glial cells, in particular astrocytes, could have an important role in it; specifically, astrocyte proteins are engaged in hydroelectrolyte balance, thus regulating the normal neuronal transmission. Between these molecules, the AQP role has been evaluated in different studies, proving its contribution in the pathophysiology of epilepsy on different levels and according to multiple mechanisms.

### 4.1. Epilepsy and AQP-Related Ions Homeostasis Mechanism

AQP4 is expressed on the plasma membrane of astrocytes and exhibits a significant concentration on the glial membrane surrounding brain capillaries. This concentration is believed to result from the anchoring of the channel through a cytoskeletal complex consisting of several proteins, including α-syntrophin, α-dystrobrevin, and dystrophin Dp71 [35,66,67]. A key role in epileptogenesis mediated by AQP4 is represented by its subcellular mislocalization; tissue swelling due to increasing extracellular volume may determine the increased concentrations of extracellular ions and neurotransmitters, intensifying neuron interactions into bursting activity [46].

AQP4 expression on astrocyte endfeet, along with Kir4.1 K+ channels, is responsible for K+ homeostasis regulation during neural activity; it was shown how an alteration of AQP4 function could affect the hyperexcitability status characteristic of epileptic brain and eventually lead to increased seizure susceptibility [18]. With regard to this concern, studies on a Kir 4.1 knockout mice model showed that alteration of K+ regulation Kir 4.1-mediated leads to the development of severe epilepsy not compatible with life and mice death a few weeks postnatally [43].

Additionally, Heuser et al. in a clinical study including 399 patients, 218 of which were affected by temporal lobe epilepsy, performing DNA sequencing on the AQP4 gene and KCNJ9 and KCNJ10 genes (coding for potassium channel Kir4.1) proved a possible association between polymorphisms of the AQP4 gene and any form of epilepsy [44].

Astrocytes play a crucial role in the pathogenesis of epilepsy modulating the expression of glutamate transporters 1 (GLT1) and AQP4. It is known that during epileptic seizures, repetitive paroxysmal electrical discharge leads to the excessive release of excitatory amino acids, such as glutamate, and certain ions, which trigger the PKA or PKC pathway via calmodulin (CaM) activation, in turn affecting the distribution of AQPs at the astrocyte level. As a matter of fact, PKA activity plays a significant role in modulating synaptic transmission and is believed to be involved in the molecular and cellular processes leading to epileptogenesis (Figure 3).

Additionally, the dysregulation of glutamate transporters, as suggested by certain authors, could contribute to the onset of epilepsy [68,69,70]. Significantly down-regulation of GLT1 e AQP4 in the post-epileptic status has been identified, leading to understand how dysregulation of these proteins’ expression induces astrocytic dysregulation of glutamate uptake and water and potassium homeostasis; increasing glutamate and potassium levels alter the synaptic microenvironment and cellular electrical potential with the result of creating a state of neuronal hyperexcitability directly related to epileptogenesis [45]. Another molecule that can interact with AQP4 and be involved in epileptogenesis is the metabotropic glutamate receptor (mGluR), a key modulator of cell excitability and synaptic transmission. Glutamate shows a role in increasing astrocyte water permeability and astrocyte swelling, and the deregulation of its receptor has been reported in epilepsy; in more detail, a lower expression was related to increased seizure frequency [71]. Moreover, AQP4 dysregulation may also be determined by impairments in astrocyte coupling enabled by the extensive gap junctions, which allows both intercellular communication and the delivery of K^+^, glutamate, and neurometabolites [46].

Intriguingly, AQP4 may also function also as a metabotropic receptor. A study by Haj-Yasein et al. found that AQP4 can interact with glutamate receptors and modulate their activity, suggesting a role in synaptic transmission and plasticity [72,73]. Indeed, using Förster resonance energy transfer (FRET) analysis, they showed that AQP4 and NMDA receptors are near each other in brain tissue; by interacting with NMDA receptors, AQP4 could regulate calcium ion influx into neurons, which is a crucial step in NMDA receptor-mediated signaling. Calcium influx through NMDA receptors plays a key role in triggering various intracellular signaling cascades that contribute to synaptic plasticity and neuronal excitability [72,74].

### 4.2. Epilepsy and AQP-Mediated Inflammation

On the hypothesis that the AQP4 expression could regulate the proinflammatory activity in the brain related to the pathogenesis of epilepsy, some studies suggest that the status of chronic and persistent brain inflammation participates in epileptogenesis or reinforces seizures [47].

Microglia are the resident immune cells in the central nervous system (CNS) and play roles in neuro-inflammation or brain injury [75]. AQP4 expression on activated microglia is related to the release of inflammatory cytokines and chemokines, such as osteopontin (OPN, an inducer of pro-inflammatory cytokine expression) and, eventually, IL-1β, TNFα, IL-6, and IL-6R. Interestingly, it has been demonstrated how in AQP4/KO mice, the induction of many inflammation- or immune system-related genes is lost [48]. Furthermore, the inhibition of AQP4 expression through acetazolamide causes a reduction of proinflammatory cytokine levels, such as IL-1b and IL-6 but not TNF-a [17]. The data strongly indicate that AQP4 plays a role in immune functions, particularly microglial activation following brain injury, at least to some extent.

Further studies are needed to elucidate the regulatory role of AQP-4 in brain inflammatory processes and to possibly confirm their involvement in the pathogenesis of seizures and epilepsy.

Riazi et al. [76] showed that a dysfunction in AQP4 homeostasis is associated with the onset of inflammatory phenomena, the result of which is increased neuronal excitability and a greater tendency for seizures.

This appears to be strongly related to the activation of microglia, with release of pro-inflammatory mediators such as tumor necrosis factor alpha (TNFα).

### 4.3. AQP, Edema, and Epilepsy 

Although the mechanism underlying epileptic seizures can be attributed to aberrant and hypersynchronous discharge of a neuronal group, laboratory studies have recently focused on the role that nonneuronal components (such as astrocytes, microglia, and endothelial cells) may play in the genesis of epilepsy [77,78]. For instance, it has been confirmed by many studies how AQP-related brain water homeostasis is involved in the etiopathology of epilepsy [49,79,80]. Particularly, AQP4 and its anchoring complex plays a key role in regulating the water permeability of the blood-brain barrier. In addition, AQP4 is involved in electrical signal propagation by facilitating the clearance of extracellular K^+^ during neuronal activity [81,82]. In this context, brain edema induces a reduction of the extracellular environment and an ion-exchange impairment, determining an abnormal spatial relationship between juxtaposed neural elements that seems to eventually promote increased seizure susceptibility [83].

Therefore, AQP4-dependent osmotic balance in astroglial cells may play an important role in neuronal activity, since AQP4-deficient mice exhibit slower K^+^ clearance and increased seizure threshold and duration. This was highlighted by the study of Kim and colleagues in which a reduction of AQP4 expression in brain tissue during the post-status epilepticus period was demonstrated, which in turn is associated with down-regulation of dystrophin and alpha-syntrophin complex. All of this results in worsening post-status epilepticus and vasogenic edema severity, determining a sort of vicious cycle [38].

Finally, studies on the role of brain edema after epileptic seizure in an AQP knockout mice model demonstrated that due to impaired clearance of tissue water, the consequent edema was greater than control group AQP+/+. The marked increase in seizure-induced tissue water content in AQP4−/− mice suggests impaired clearance of tissue water. In addition, previous research supports a transient increase in water content following status epilepticus [84].

Considering previous findings on disruption of the BBB during intense seizure activity, it has been hypothesized that AQP4 probably plays a crucial role in the elimination of excess water entering the brain due to disruption of the blood-brain barrier during an episode of status epilepticus. Interestingly, the very presence of edema may contribute to the development of epilepsy, as decreased osmolarity and reduced extracellular space volume can have a significant impact on excitability [85,86,87,88]. Undoubtedly, further studies are needed to better define the relationship between seizures, edema, and AQP4 dysfunction.

### 4.4. AQP, Trauma, and Epilepsy

Several studies showed a possible role of AQP in trauma-related epilepsy [51,52,89].

Lu et al. [52], in their study, have proposed that dysregulation in AQP4 leads to post-traumatic seizure vulnerability, promoting microgliosis. Additionally, their research reveals that the prolonged and heightened presence of microglia may serve as a pathogenic mechanism for the inflammatory condition, potentially affecting seizure susceptibility. This mechanism is further supported by the observation that inhibiting microglia with minocycline reverses the seizure phenotype in AQP4-null mice. These observations imply that glial scar formation, through a microglia-dependent process, is directly related to post-traumatic seizure development. These findings may have relevance not only in traumatic injury scenarios but also in the surgical management of central nervous system disorders, where surgical trauma occurs and postoperative antiepileptic drugs are utilized [90,91,92,93].

Undoubtedly, further studies, possibly in human models, are needed for a better understanding and characterization of this evidence.

It was also observed that the increased permeability of the blood-brain barrier (BBB) following brain insults, in terms of BBB leakage, is induced during post-epilepticus in the main epileptogenesis-associated brain regions, eventually creating new pro-epileptic consequences [94]. A reduction in AQP4 expression in areas of BBB leakage has been shown, indicating a severe disturbance of astrocyte-mediated endothelial-neuronal coupling, promoting epileptogenesis [50]. Studies evaluating AQP4 expression in both the frontal cortex and the hippocampus on a mice model with post-traumatic epilepsy demonstrated that AQP4 levels were higher than in mice that did not develop post-traumatic epilepsy. Indeed, AQP4 dysregulation after brain injury, in terms of upregulation and mislocalization of perivascular AQP4, is the primary risk factor in the development of posttraumatic epileptogenesis due to the resulting impaired water and ion homeostasis and, eventually, hyperexcitability [51].

Finally, studies on post-traumatic epilepsy using an AQP4 knock-out mice model showed that AQP4 seems to play a protective role in post-traumatic seizures by promoting astrogliosis, forming a glial scar, and preventing microgliosis. AQP4 KO-mice had a shortened seizure latency and increased seizure severity in association with increased microglial activity in comparison to the control group, where, through astrocyte scar-formation activity, post-traumatic seizures were reduced [95].

### 4.5. AQP4, Mesial Temporal Lobe Epilepsy and Drug-Resistant Epilepsy

Mesial temporal lobe epilepsy (MTLE) is one of the most common types of focal epilepsy in the adult population [96]. It is a chronic seizure disorder that is often refractory to medical treatment [97,98]. In addition, in this regard, several studies have evaluated the physiopathological role of AQP4.

Interesting are the results of Salman and colleagues. They directed a clinical study on patients with pharmaco-resistant MTLE associated with unilateral hippocampal sclerosis [53]. ELISA data on post-amygdalohippocampectomy samples highlighted a remarkable rise in AQP4 protein expression in sclerotic tissue samples compared to the non-sclerotic samples. Intriguingly, they also pointed to the role of a MAPK-signaling AQP4-expression regulation pathway in temporal lobe epilepsy pathogenesis. 

In this regard, previous studies demonstrated how hyperosmotic dehydration represents a trigger point of MAPK pathways activation, with a subsequent increase in AQP expression in astrocytes [54]. Indeed, AQP4 expression is higher in the sclerotic hippocampus (CA1 region), suggesting the role of this water channel in the pathogenesis of intractable epilepsy [55,99].

In the same way, Zhou et al. have examined the expression of AQP1 in surgical samples of the anterior temporal neocortex of patients with intractable epilepsy, showing a relationship between AQP1 and forms of intractable epilepsy. Their findings have shown increased expression of AQP1 at the astrocyte level, absent in the neuronal and oligodendroglial components; their data support the idea that altered homeostasis of water and electrolytes, particularly K^+^, underlies some forms of intractable epilepsy. They pointed out that during epileptic seizures, K^+^ is released from the active neuropil into the extracellular space, where it is eliminated by the ends of astrocytes, which cause osmotic swelling. The swelling of astrocytes restricts the extracellular space, increasing epileptiform activity [100]. Furthermore, alterations in AQP1 would appear to promote glial scar formation, which in turn would produce deleterious effects, including defects in neuronal transmission, alteration of neural network homeostasis, and the onset of drug-resistant epilepsy [6,101,102].

Previous data suggested that AQP dysregulation effects on intractable epilepsy could be related to the expression of multidrug resistant protein 1 (MRP1) and P-glycoprotein (Pgp) [103,104]. Duan et al. showed, on mice affected by MTLE, that acetazolamide (AQP4 inhibitor) causes a reduction of multidrug resistant proteins, highlighting a useful therapeutic checkpoint for intractable epilepsy [39].

Ultimately, Gondo et al. illustrated, in an in vitro model of mice-derived cortical brain slices, that the continuous epileptic activation of neurons decreases the expression level of beta-dystroglycan (beta-DG) due to a downregulation of AQP4 on astrocyte endfeet, suggesting that the apparent recovery of neuronal activities (EEG finding) does not represent the recovery of the brain tissue [56,105]. All these data are worthy of focused investigation, especially to establish new pharmacological options.

### 4.6. AQP and Tumor-Associated Epilepsy

One of the most common symptoms in patients with brain tumors is epilepsy [106,107,108]. In this context, seizures occur as the initial symptom in 20–40% of patients, and an additional 20–45% of patients will experience them at some point during the course of the disease. The overall occurrence rate of epilepsy in neuro-oncology patients, irrespective of the specific histological type and anatomical location of the lesion, ranges from 35 to 70% [109,110,111]. In addition to being a terrible burden for the patient, severely affecting his or her neurocognitive state, it represents a complex therapeutic profile and requires a unique and multidisciplinary approach. These patients, in fact, must face two different pathologies at the same time, brain tumor and epilepsy [112,113]. One of the tumor histotypes most frequently associated with the occurrence of seizures is glioblastoma (GB) [114]. Even if the exact mechanisms involved in GB-related epilepsy are still poorly understood, several theories propose various factors contributing to the development of epilepsy in this setting. These factors include peritumoral amino acid imbalances, local metabolic alterations, cerebral edema, pH abnormalities, morphological alterations of the neuropil, changes in the expression of neuronal and glial enzymes and proteins, and altered immunologic activity [115,116,117]. The occurrence of seizures can complicate the clinical course of patients with GB. In addition, about half of GB patients experience at least one seizure during their disease [118,119]. A putative role in the multifactorial genesis of GB-related epilepsy may be sought in dysregulation of AQP4. As demonstrated by Isoardo and colleagues [57], AQP4 dysregulation could be involved in the onset of seizures in patients affected by GB. Interestingly, their study highlighted that reduced expression of AQP4 is associated with a reduced risk of seizures in patients with GB. Furthermore, they have found that reduced expression of cell surface AQP4 is characteristic of GB patients without seizures, and this feature could be likely attributable to a posttranslational mechanism. Other studies have pointed out that dysregulation of AQP4 and other related proteins and alteration of the BBB always resulting from malfunction of the aquaporin system are implicated in GB-related epileptogenesis [120,121,122,123]. Further studies are obviously needed to substantiate these hypotheses and possibly develop new therapeutic strategies.

### 4.7. Aquaporins as Potential Therapeutic Target in Brain Disease

AQPs have emerged as promising therapeutic targets for brain disorders, including diseases that predispose to seizure onset. As mentioned above, these proteins play an important role in controlling water transport and ion homeostasis in brain cell membranes. Abnormal fluid accumulation can occur in various neurological conditions such as NMO [124], traumatic brain injury [125], hydrocephalus [126], and ischemic stroke [12], resulting in brain inflammation and predisposition to epileptogenesis. In this context, the possibility of targeting aquaporins to stimulate physiological fluid flow and restore the balance of brain fluid homeostasis has gained interest in clinical settings. Researchers have sought to develop specific aquaporin inhibitors or activators that could selectively regulate fluid flow in the brain [127,128,129]. For instance, Sun and colleagues have demonstrated how acute inhibition of AQP4 by TGN-020 was found to improve neuronal recovery and disease outcomes by reducing cerebral edema and peri-infarct astrogliosis [130]. However, to date there are still some limitations in the development of modulators of the activity of AQPs, especially AQP4. Major challenges in identification and clinical development include the numerous homologous isoforms of AQPs with wide distribution and tissue functions, the undesired actions expected at the CNS and non-CNS levels, and the need for high blood-brain barrier permeation [131]. Undoubtedly, further investigations will be essential to overcome these obstacles.

## 5. Conclusions

As neurosurgeons, we are confronted daily with cases of epilepsy, both idiopathic and secondary to well-evident pathological processes. We are aware of the importance of understanding in detail the mechanisms underlying epileptogenesis to find viable therapeutic strategies to control epilepsy, prevent its recurrence, and improve the quality of life of affected patients. However, the pathogenetic basis of epilepsy is still debated today. A growing body of evidence indicates that aquaporin proteins, particularly AQP4, play a key role in the process of epileptogenesis by mediating water transport in the brain, maintaining K+ homeostasis and proper extracellular volume, ensuring proper BBB function, acting as modulators of inflammation, and interacting with synaptic normofunction. However, other actors besides AQPs seem to play key roles in the pathogenesis of epilepsy, from the anchoring system of AQP4 itself to the phosphorylation state of some protein components to genetic and post-transcriptional regulatory mechanisms.

Over the last few years, many efforts have been made to pinpoint the multiple factors involved in the genesis of epilepsy, thus overlooking the specific role of aquaporins.

This review aims to highlight the available data in this regard in a way that is easy and accessible to all, also laying the foundation for future studies and shedding light on the search for possible therapeutic options targeting the aquaporin family. Much remains to be discovered or better stated; certainly, the increasing understanding of the molecular mechanisms underlying epilepsy and the study of various mediators represent an effort in the discovery of new fascinating therapeutic horizons.

## Figures and Tables

**Figure 1 ijms-24-11923-f001:**
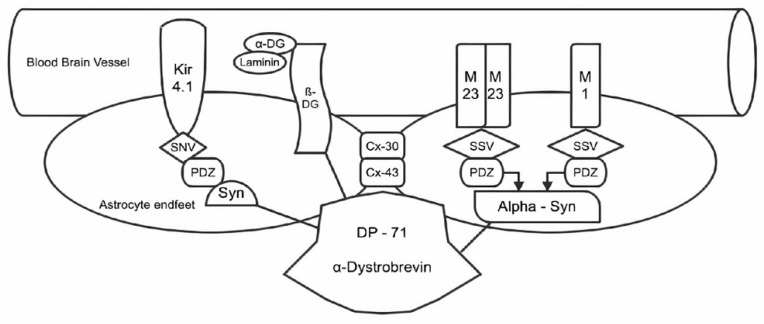
Anchoring system of AQP4 in CNS. The dystrophin complex is connected to the basal lamina, and it interacts with AQP4 via α-syntrophin (α-syn) or other syntrophins (syn). It is hypothesized that α-syn plays a crucial role in anchoring the M23 isoform of AQP4, which is responsible for forming orthogonal arrays of proteins, particularly concentrated in endfeet membranes.

**Figure 2 ijms-24-11923-f002:**
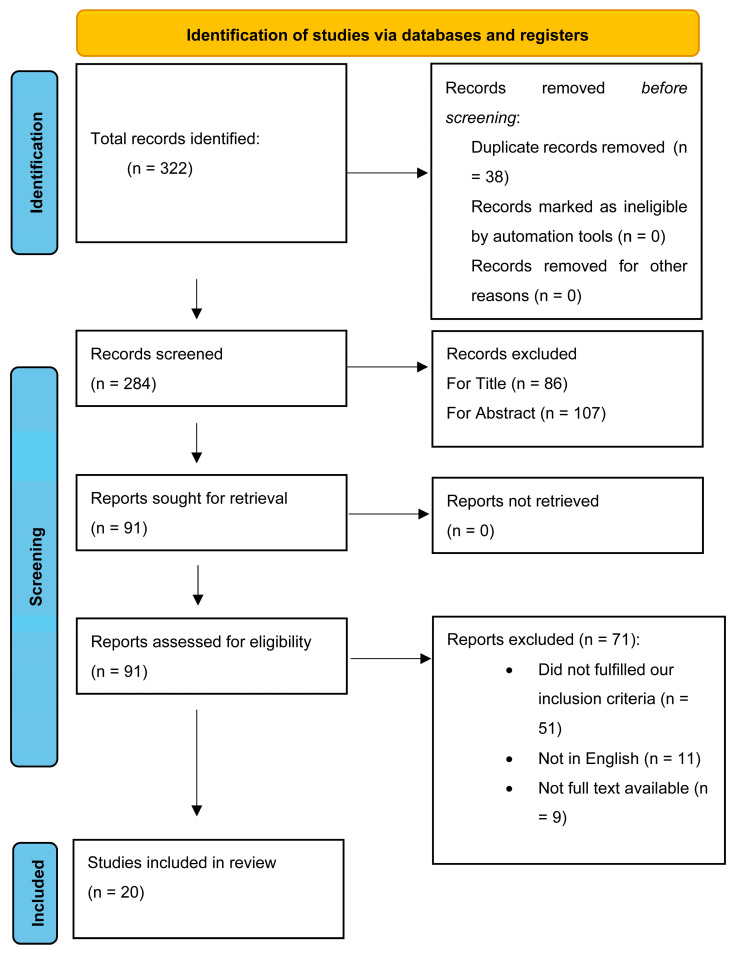
PRISMA flow chart.

**Figure 3 ijms-24-11923-f003:**
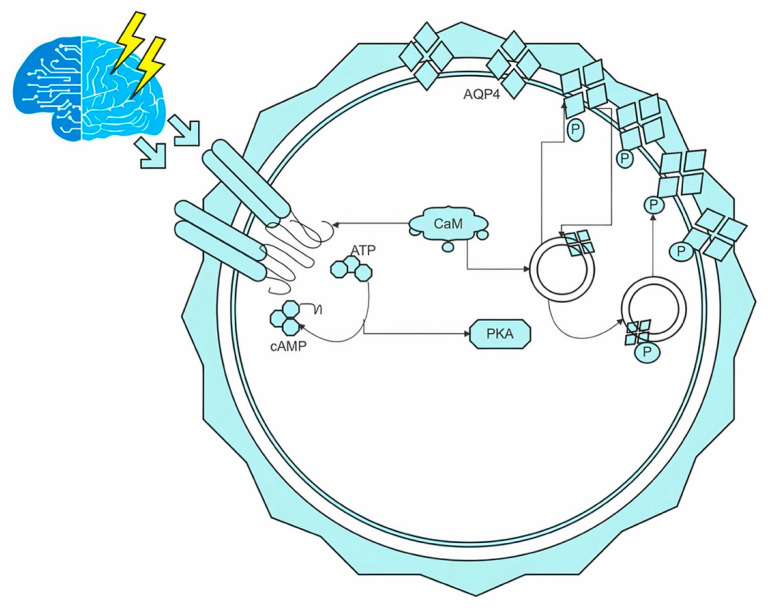
In stress conditions, such as after an injury (hypoxic insult), astrocyte CaM is activated and interacts with adenylyl cyclase, activating in turn cyclic AMP (cAMP)-dependent PKA, which phosphorylates AQP4 at Ser276, determining its displacement to the plasma membrane. CaM interacts directly with AQP4, driving AQP4 to subcellular mislocalization and disrupting its physiological homeostasis.

**Table 1 ijms-24-11923-t001:** List of human aquaporins and their location and roles in the organism.

Aquaporin	Sites	Functions
AQP0	Lens	Cataract (Loss of Function)
AQP1	Epithelial cells of choroid plexus, inner ear, spinal cord, renal tubules, red blood cells, dermis, blood vessels, liver, pancreas, articular cartilage, intervertebral disc	Water and ion balance, cerebrospinal fluid secretion
AQP2	Renal collecting duct	Urine concentration Nephrogenic diabetes insipidus (loss of function)
AQP3	Renal collecting duct, adipocytes, epidermis, airway, gastrointestinal tract, articular cartilage, intervertebral disc, female and male reproductive system	Glycerol permeability, urine concentration, water balance in pregnancy and parturition and normal fertility
AQP4	Astrocytes, retina, olfactory epithelium, inner ear, spinal cord, cardiomyocites, airway, small intestine, muscle fibers	Cerebrospinal fluid flux, transcellular water flow CNS diseases (loss of function)
AQP5	Glandular tissues, lung and airway	Saliva secretion, sweat excretion, transcelluar water flow
AQP6	Renal collecting duct	Acid secretion
AQP7	Adipocytes, renal tubule, ovarian granulosa cells, sperm, gastrointestinal tract	Glycerol permeability, transcellular water flow, sperm maturation
AQP8	Intestinal epithelial cells, spinal cord, ovarian granulosa cells	Transcellular water flow
AQP9	Hepatocytes, red blood cells, brain, female and male reproductive system, eye, ear, osteoclasts, gastrointestinal tract	Glycerol permeability, osteoclast differentiation, glucose cell metabolysm
AQP10	Adipose tissue, gastrointestinal tract, male reproductive system, skin	Glycerol efflux, skin barrier, water balance
AQP11	Liver, testis	Not identified
AQP12	Exocrine pancreas	Not identified

**Table 2 ijms-24-11923-t002:** Aim and major findings of studies included in the review.

Author	Year	Aim of the Study	Principal Findings
*AQPs & Ions Homeostasis*			
Djukic et al. [43]	2007	To study the role of the Kir 4.1 channel in a Kir4.1 cKO epilepsy mice model	The impairment of astrocyte K+ and glutamate uptake, induced by the loss of Kir4.1, affects neuronal functioning by decreasing neuronal spontaneous activity and enhancing synaptic potentiation
Heuser et al. [44]	2010	To identify variants of AQP4 and KCNJ9 genes associated with TLE and other TLE subgroups entities	Polymorphism in AQP4 and KCNJ10/KCNJ9 genes seems to be associated with temporal lobe epilepsy
Binder et al. [18]	2012	To highlight the role of AQP in the pathogenesis of epilepsy	AQPs have a crucial role in brain water and K+ homeostasis, which have powerful effects on seizure susceptibility
Hubbard et al. [45]	2016	To examine GFAP, GLT1, and AQP4 regulation during epileptogenesis in a temporal lobe epilepsy mouse model	GLT1 and AQP4 are downregulated during the early epileptogenic period, associated with increased glial fibrillary acidic protein (GFAP) expression; early disruption of astrocytic water, potassium, and glutamate homeostasis could have powerful epileptogenic and cognitive effects
Szu et al. [46]	2022	To evaluate the role of AQP4 dysregulation and mislocalization in epilepsy	Volume changes in the extra-cellular space have a relevant influence on neuronal activity and are regulated by AQP4, whose mislocalization represents a key element in epilepsy generation. Moreover, a few molecules that are thought to interact with AQP4 must be considered, such as Kir4.1, GLT-1 and mGluR5
*AQPs & Inflammation*			
Vezzani et al. [47]	2008	To collect several studies about the role of inflammatory cytokines in the pathogenesis of epilepsy	The study points to novel glio-neuronal communications in diseased conditions, highlighting potential new targets for therapeutic intervention
Ikeshima-Kataoka et al. [48]	2013	To examine the proteic expression profiles in wild-type (WT) and AQP4-deficient (AQP4/KO) mice model after an induced brain stab injury	AQP4 is clearly implicated in the induction of osteopontin (OPN) as well as pro-inflammatory cytokines in injured mouse brain
Yu et al. [17]	2016	To demonstrate the role of AQP4 in the regulation of proinflammatory cytokine expression and their effect in chronic epilepsy mice model.	AQP4 inhibition could weaken excitotoxicity in epileptogenesis by reducing proinflammatory cytokines in the hippocampus
*AQPs & Edema*			
Kim et al. [38]	2010	To analyze the role of AQP4 in brain edema formation following status epilepticus in a pilocarpine-induced rat epilepsy model	Status epilepticus (SE) may induce impairments of astroglial AQP4 functions via disruption of the dystrophin/a-syntrophin complex, worsening vasogenic edema.
Lee et al. [49]	2012	To investigate the role of AP4 in the development of cerebral edema following kainic acid-induced status epilepticus (SE)	The results indicate significantly greater tissue edema and T2 MRI changes in AQP4−/−compared to AQP4+/+mice that peaks at about 1 day after SE (greater in hippocampus relative to cortex)
*AQPs & Trauma*			
Bankstahl et al. [50]	2018	To provide comprehensive data on the spatiotemporal evolution of status epilepticus-induced BBB leakage in vivo and ex vivo in an epilepsy rat model	Prompt BBB-stabilizing intervention is necessary to prevent distinct albumin extravasation and subsequent pro-epileptogenic consequences following SE
Szu et al. [51]	2019	To characterize changes in aquaporin-4 (AQP4) expression in subjects with post-traumatic epilepsy (PTE)	The findings indicate a notable elevation in the level of AQP4 in the frontal cortex and hippocampus of mice that experienced post-traumatic epilepsy (PTE) compared to those without PTE. Additionally, in mice with PTE, AQP4 seemed to be relocated from the perivascular endfeet to the neuropil region.
Lu et al. [52]	2021	To examine the differences of post-traumatic seizure susceptibility between AQP4-deficient mice (AQP4−/−) after injection of pentylenetetrazole (PTZ) 1 month after controlled cortical impact (CCI) and wild-type sham injury control mice	Protective role of AQP4 in post-traumatic seizure susceptibility by promoting astrogliosis, formation of glial scar, and prevention of microgliosis
*AQPs, Temporal Lobe Epilepsy (TLE) & Drug resistance*			
Salman et al. [53]	2017	To analyze mRNA and protein expression of human cerebral AQPs in sclerotic hippocampi (TLE-HS) and adjacent neocortex tissue (TLE-NC) of TLE patients.	Increased AQP4 protein expression in TLE-HS samples compared to TLE-NC. Moreover, the transcript expression of AQPs 1 and 4, astrocytic biomarker and glial fibrillary acidic protein, was significantly increased, while the transcript expression of AQP9 significantly reduced in TLE-HS compared to TLE-NC. There was no significant difference in mRNA levels for AQP3, AQP5, AQP8, AQP11 or Kir4.1.
Yang et al. [54]	2013	To investigate AQP expressions in astrocytes under hyperosmotic stress, as well as the role of MAPKs in the AQP expressions. Moreover, the role of AQP3, AQP5, and AQP8 in astrocyte water movement was also studied	AQP4 and AQP9 expression continuously increased until 12 h after hyperosmotic solution exposure, whereas AQP3, -5, and -8 expression increased until 6 h under the same conditions
Lee et al. [55]	2004	To study the expression of aquaporin-4 (AQP-4) in human sclerotic hippocampus affected by temporal lobe epilepsy.	Altered expression of AQP-4 and dystrophin could potentially contribute to the disruption of ion and water balance in the sclerotic hippocampus, ultimately playing a role in the epileptogenic characteristics of the sclerotic tissue.
Zhou et al. [6]	2008	To investigate the role of AQP1 in pathophysiology of intractable epilepsy	Overexpression of AQP1 is correlated to extracellular space reduction which triggers the seizure activity.
Lei Duan et al. [39]	2017	To investigate the effects of aquaporins (AQP) inhibitor on multi-drug-resistant protein expression in an MTLE rat model.	Acetazolamide, AQP4 inhibitor, could reduce the expression of multi-drug-resistant proteins, such as MRP1 and Pgp
Gondo et al. [56]	2014	To examine the potential molecular changes of cortical astrocytes on their endfeet.	Continuous epileptic activation of neurons for 1 h decreases the expression level of dystroglycan (DG) in acute cortical brain slices prepared from mice.
*AQPs and GBM-related seizures*			
Isoardo et al. [57]	2012	To investigate the relationship between the AQP-4 expression in patients with GBM and the occurrence of seizures.	Reduced expression of AQP-4, probably related to posttranslational mechanisms, is associated with a reduced risk of seizures in patients with GBM

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
