# Peer review of "The Role of Aquaporins in Epileptogenesis—A Systematic Review"

_ijms, 2023, doi:10.3390/ijms241511923_

Round 1

Reviewer 1 Report

Title: "The Role of Aquaporins in Epileptogenesis. A Systematic Review"

Abstract: "Abstract: Aquaporins (AQPs) are a family of membrane proteins involved in the transport of water 13
and ion across cell membranes. Epileptogenesis is a complex and multifactorial process that 14
involves alterations in the structure and function of neuronal networks. AQPs have been shown to 15
be implied in various physiological and pathological processes in the brain, including water 16
homeostasis, cell migration, and inflammation among others. Recent evidence suggests that AQPs 17
may also play a role in the pathogenesis of epilepsy. In animal models of epilepsy, AQPs have been 18
shown to be upregulated in regions of the brain that are involved in seizure generation, suggesting 19
that they may contribute to the hyperexcitability of neuronal networks. Moreover, genetic studies 20
have identified mutations in AQP genes associated with an increased risk of developing epilepsy.. 21
The results suggest that AQPs could serve as a promising treatment target for epilepsy in the coming 22
years. Our review aims to investigate the role of AQPs in epilepsy and seizure onset from a 23
pathophysiological point of view. "

General comment. Altough the topic of the review is related to the special issue, this manuscript should be deeply reworked to enhance its quality and impact.
It seems that some important informtation is lacking or is not properly provided as AQP structure, Sequence and structural features of AQPs, Aquaporin expression in the nervous system,
AQP functions in the nervous system, etc,etc...

In particular the following major points prevent this work from the publication:

1) "introduction" section is too small for a review. Some important works related to Aquaporins seem to be neglected. Please improve this section.
2) "2. Materials and Methods" section. It is not clear the importance of the description of the "PRISMA flow chart"
3) "3. Results " also this section is not clear. Please explain in a better way.
4) "4. Discussion" in this section some subsections should be moved in other parts of the manuscript: e.g.,Abbreviation: -DG: alpha-Dystroglycan; -DG: beta- Dystroglycan; Kir 4.1: Inward-rectifier 194
potassium channels 4.1; Cx-30/Cx-43: Connexin 30 and Connexin 43; -syn: Alpha-syntrophin; 195
M23/M1: AQP4 isoform; SSV/SNV/PDZ: C-terminal 3 amino acids, −SSV/SNV, serve as the ligand 196
of a PDZ binding partner; DP-71: Dystrophin 71. 197
Figure 2 and the caption,
Fig. 2.: Anchoring system of AQP4 in CNS. The dystrophin complex is connected to the basal lamina, 198
and it interacts with AQP4 (as well as potentially other molecules containing a C-terminal SXV 199
sequence, such as Kir4.1) via α-syntrophin (α-syn) or other syntrophins (syn). It is hypothesized that 200
α-syn plays a crucial role in anchoring the M23 isoform of AQP4, which is responsible for forming 201
orthogonal arrays of proteins, particularly concentrated in endfeet membranes.

"Abbreviation: AQP4: aquaporin 4; P: Phosphate; CaM: Calmodulin; PKA: protein kinase A; ATP: 224
adenosine triphosphate; cAMP: Cyclic adenosine monophosphate 225
Fig. 3.: In stress condition or after an injury (hypoxic insult) astrocytes CaM is activated and interacts 226
with an adenylyl cyclase, activating in turn cyclic AMP (cAMP)-dependent PKA, which 227
phosphorylates AQP4 at Ser276, causing it to relocalize to the plasma membrane. CaM interacts 228
directly with AQP4, driving AQP4 to subcellular relocalization. "

All figures should be provided in high resolution and improved as well as their captions

The language could be improved

Author Response

Dear reviewer,

I hope this email finds you well.

First, I would really thank you for your time and your kind suggestions.

In accordance with your advice, we implemented the introductory section, trying to clarify the anchoring system of AQP4 to astrocytic perivascular end feets and the other proteins involved in this system. In this regard, we have moved Figure 2 to this section.

We then proceeded to improve both the materials and methods section, explaining more clearly the process of study selection according to the PRISMA flow chart, and the results.

Finally, we tried to improve the resolution of the images and their captions. We have then checked the English language.

We hope to have addressed to your tips in the best way.

Sincerely,

Lapo Bonosi

Reviewer 2 Report

This is very interesting work. I have two suggestions:

  1. The paragraph describing AQPs as therapeutic targets should be included.
  2. The figures should be improved, especially Figure 1. It is very hard to understand their message.

The quality of the English is okay. As a non-native speaker, I cannot see significant errors, and I have no problems understanding the text.

Author Response

Dear reviewer,

Many thanks for your time and suggestion

We really appreciate your advice, and we modified the manuscript according to it. We have tried to clarify the image 1, changing the caption text, and delving into the anchoring system of AQP4 to astrocytes in the introduction. We then moved the figure 1 in introduction section. Furthermore, we have added a new paragraph regarding aquaporins as potential therapeutic target in the discussion section. We have voluntarily decided not to delve too deeply into the aspect inherent in the development of potential drugs modulating the activity of AQPs as we consider the topic complex and needing of another article to better explore it. However, we consider it appropriate to emphasize the fact that much research is focused on this aspect, and we are confident that new therapeutic options will become available in the coming years. Finally, we have improved the resolution of the images and their captions.

Sincerely,

Lapo Bonosi

Reviewer 3 Report

The manuscript by Bonosi et al. presents a review on the role of aquaporins (APs) in epileptic pathogenesis. The authors have analyzed the literature on APs functioning with the use of PRISMA flowchart strategy and selected the most important papers dealing with the APs in relation to epilepsy. Possible mechanisms of this pathology were elucidated. It was shown that APs can be promising drug targets in the epilepsy treatment. This analytical review is important and deserves publication.

Specific comments:

Line 40:  Please replace "have been describes" with "have been described".

Line 53:  The abbreviation "CM" appears here for the first (and single) time in the manuscript and should be explained.

Line 103:  Regarding the PRISMA flowchart, literature reference(s) are needed defining or describing main principles of this flowchart.

Line 234:  Please check the phrase "GLT1 e AQP4" (Italian?).

I recommend acceptance of the manuscript for publication after minor revision.

Author Response

Dear reviewer,

Many thanks for your time and suggestions

We are happy that you enjoyed the manuscript. We have made the corrections you indicated.

Best regards,

Lapo Bonosi

Round 2

Reviewer 1 Report

Title: "The Role of Aquaporins in Epileptogenesis. A Systematic Review"

Abstract: "Abstract: Aquaporins (AQPs) are a family of membrane proteins involved in the transport of water and ion across cell membranes. Epileptogenesis is a complex and multifactorial process that involves alterations in the structure and function of neuronal networks. AQPs have been shown to be implied in various physiological and pathological processes in the brain, including water homeostasis, cell migration, and inflammation among others. Recent evidence suggests that AQPs may also play a role in the pathogenesis of epilepsy. In animal models of epilepsy, AQPs have been shown to be upregulated in regions of the brain that are involved in seizure generation, suggesting that they may contribute to the hyperexcitability of neuronal networks. Moreover, genetic studies have identified mutations in AQP genes associated with an increased risk of developing epilepsy. The results suggest that AQPs could serve as a promising treatment target for epilepsy in the coming years. Our review aims to investigate the role of AQPs in epilepsy and seizure onset from a pathophysiological point of view. "

General comment. Altough the authors tryied to review their previous version of the manuscript, unfortunately, the main text is almost the same. Therefore, also the quality and the major points to review are more or less the same. Indeed, as already said in the previous round of revision, this (review) work should provide more detailed information to the interested readers about:

Aquaporins(-4) identification, distribution, structure and function

Sequence and structural features of AQPs

AQP4 assembly in orthogonal arrays of particles

Functions of AQP4 in the CNS

Possible prospects for AQP4-based therapies etc.

In addition, the value of this work is not clear with reference to the previous literature.  Some examples:

Vandebroek, A.; Yasui, M. Regulation of AQP4 in the Central Nervous System. Int. J. Mol. Sci. 2020, 21, 1603. https://doi.org/10.3390/ijms21051603

Mechanisms Underlying Aquaporin-4 Subcellular Mislocalization in Epilepsy, Front. Cell. Neurosci., 06 June 2022, Sec. Cellular Neuropathology , Volume 16 - 2022 | https://doi.org/10.3389/fncel.2022.900588

Regulation of Brain Water: Is There a Role for Aquaporins in Epilepsy? F Edward Dudek, Ph.D. and Michael A Rogawski, M.D., Ph.D.,Epilepsy Curr. 2005 May; 5(3): 104–106.doi: 10.1111/j.1535-7511.2005.05310.x

Aquaporin-4 and epilepsy, Devin K Binder 1 , Erlend A Nagelhus, Ole Petter Ottersen, Glia. 2012 Aug;60(8):1203-14. doi: 10.1002/glia.22317. Epub 2012 Feb 29.

Brittany Short, Lindsay Kozek, Hannah Harmsen, Bo Zhang, Michael Wong, Kevin C. Ess, Cary Fu, Robert Naftel, Matthew M. Pearson, Robert P. Carson, Cerebral aquaporin-4 expression is independent of seizures in tuberous sclerosis complex, Neurobiology of Disease, Volume 129, 2019,Pages 93-101, ISSN 0969-9961,https://doi.org/10.1016/j.nbd.2019.05.003.

As a consequence, the authors should further improve the quality of this work providing an original point of view of the problem, which could support the need of this manuscript.

The quality of the language could be futher improved

Author Response

Dear Reviewer,

Thank you again for your time and your suggestions.

We modified the manuscript adding new information regarding the molecular structure and sequence of AQP4, hoping to better define its importance also with a view to finding possible treatment options.

Regarding your general comment “Although the authors tried to review their previous version of the manuscript, unfortunately, the main text is almost the same. Therefore, also the quality and the major points to review are more or less the same”, we modified the manuscript providing a more detailed explanation of the search strategy and process, changing both materials&methods and results section. So, the main text is not “almost the same” from our point of view.  

About Aquaporins (-4) identification, distribution, and function, in the introduction we briefly explained the discovery of aquaporins family, their distribution in human body and the various function performed by each of them as one can see in table 1. We decided not to go into too much detail regarding the structure and specific functions of each isoform of AQPs as it is not strictly relevant to the objective of our review and moreover already covered in more detail in other works.

Regarding possible prospects for AQP4-based therapies, we added a subsection in the discussion in which we also tried to give the reader some information on the state of the art of possible modulatory therapies on the activity of AQPs, particularly isoform 4. However, we consider the focus worthy of further investigations in a separate paper and not in this review.

Sincerely

Lapo Bonosi

Round 3

Reviewer 1 Report

Title: "The Role of Aquaporins in Epileptogenesis. A Systematic Review"

Abstract: “ Aquaporins (AQPs) are a family of membrane proteins involved in the transport of water and ion across cell membranes. Epileptogenesis is a complex and multifactorial process that involves alterations in the structure and function of neuronal networks. AQPs have been shown to be implied in various physiological and pathological processes in the brain, including water homeostasis, cell migration, and inflammation among others. Recent evidence suggests that AQPs may also play a role in the pathogenesis of epilepsy. In animal models of epilepsy, AQPs have been shown to be upregulated in regions of the brain that are involved in seizure generation, suggesting that they may contribute to the hyperexcitability of neuronal networks. Moreover, genetic studies have identified mutations in AQP genes associated with an increased risk of developing epilepsy. The results suggest that AQPs could serve as a promising treatment target for epilepsy in the coming years. Our review aims to investigate the role of AQPs in epilepsy and seizure onset from a pathophysiological point of view. "

General comment: The authors revised their work and this is laudable. The revised version, which is downloadable from the site is suboptimal, since it is not too clear. However, although I am not totally convinced of the need of this work, from a technical point of view this could be a publishable work after the following minor revision:

*) Please, rewrite the “Conclusion” section, underlining in a better way the value of this work with respect to the current state of the art.

The quality of the language could be improved

Author Response

Dear Reviewer,

Thank you again for your comment.

We modified the manuscript remark in the conclusion what is it for us the aim of this review and because it is important and valuable for scientific community.

As neurosurgeons, we are often confronted with epilepsy and pathologies that result in epileptogenesis. And yet the mechanisms underlying this pathology are still unclear. We thought that a review conducted in this way might offer an opportunity even to those who do not deal with this topic to still be able to gain insight into the extreme complication behind this disease, trying in our own small way to shed light on the multiple actors involved.

Best regards,

Lapo Bonosi
